# Effects of an anti-lipogenic low-carbohydrate high polyunsaturated fat diet or a healthy Nordic diet versus usual care on liver fat and cardiometabolic disorders in type 2 diabetes or prediabetes: a randomized controlled trial (NAFLDiet)

Polyunsaturated fatty acids (PUFA) have been shown to reduce liver fat compared to saturated fat, but effects of a novel "anti-lipogenic" diet replacing carbohydrates with PUFA (LCPUFA) or a low-fat healthy Nordic diet (HND) rich in whole-grains are unknown. The objective of this study was to investigate the effects of these diets, as compared with usual care (UC), on liver fat (primary outcome) and related glycemic and lipid disorders after 12 months of intervention, in individuals with prediabetes or type 2 diabetes (T2D). A three-arm parallel ad libitum randomized trial was completed in December 2022 (NCT04527965). Outcome assessors and care providers were blinded to participants' diets. Men and women (n=150) with prediabetes or T2D (55%) were randomized in a 1:1:1 allocation ratio, stratified by sex and T2D status, and were assessed at the Uppsala Academic hospital. General linear models were employed to estimate intention-to-treat effects. Liver fat was reduced after the LCPUFA diet (n=54) and the HND (n=51) when compared to UC (n=43); -1.46% (95% CI: -2.42, -0.51)) and -1.76 % (95% CI: -2.96, -0.57), respectively. No difference in liver fat between LCPUFA and HND was observed. Body weight and HbA1c decreased more in the HND versus the other diets, whereas no differences were observed between LCPUFA and UC. LDL-cholesterol was reduced to a similar extent during the HND and LCPUFA diet, compared to UC, whereas only HND reduced triglycerides, inflammation and liver enzymes. In total, n=4 serious adverse events occurred, distributed among groups. An ad libitum mainly plant-based LCPUFA diet and HND similarly reduced liver fat and LDL-cholesterol, compared with UC. Even without intentional energy restriction, the HND further improved body weight, glycemic control, liver biochemistry, triglycerides, and inflammation, suggesting a HND as a clinically feasible diet for the management of T2D and metabolic dysfunction-associated steatotic liver disease (MASLD).

e-mail: ulf.riserus@uu.se

**Table 1 | Baseline characteristics[a]**

| | LCPUFA (n = 54) | HND (n = 51) | UC (n = 43) |
|---|---|---|---|
| Sex (n (%) women) | 22 (41) | 20 (39) | 16 (37) |
| Age (years) | 65 (10) | 65 (10) | 65 (10) |
| BMI (kg/m²) | 30.1 ± 3.6 | 29.5 ± 3.7 | 30.3 ± 3.2 |
| Current smoking (n (%)) | 6 (11) | 1 (2) | 2 (5) |
| University (n (%)) | 34 (63) | 29 (57) | 30 (70) |
| T2D (n (%)) | 30 (56) | 23 (45) | 28 (65) |
| Alcohol consumers (n (%)) | 44 (81) | 47 (92) | 38 (88) |
| PEth (µmol/L) | 0.05 (0.06) | 0.05 (0.05) | 0.05 (0.02) |
| Liver fat (%) | 6.3 (6.9) | 6.4 (5.5) | 8.7 (10.8) |
| NAFLD (n (%)) | 26 (48) | 30 (59) | 30 (70) |
| Glucose (mmol/L) | 6.7 (1.9) | 6.7 (1.6) | 6.7 (1.0) |
| HbA1c (mmol/L) | 42.0 (12.0) | 40.0 (10.0) | 40.0 (9.0) |
| Total cholesterol (mmol/L) | 4.6 ± 1.2 | 5.1 ± 1.3 | 5.0 ± 1.1 |
| LDL-cholesterol (mmol/L) | 2.8 ± 1.1 | 3.2 ± 1.1 | 3.1 ± 1.1 |
| Triglycerides (mmol/L) | 1.3 (0.6) | 1.3 (0.8) | 1.4 (0.8) |
| SBP (mmHg) | 130.6 ± 14.6 | 134.3 ± 14.3 | 135.4 ± 13.5 |
| DBP (mmHg) | 83.1 ± 9.0 | 84.8 ± 8.2 | 85.4 ± 8.3 |
| Antidiabetic medications (n (%)) | 22 (41) | 20 (39) | 24 (56) |
| Antihypertensive medications (n (%)) | 34 (63) | 25 (49) | 25 (58) |
| Lipid-lowering medications (n (%)) | 26 (48) | 21 (41) | 15 (35) |
| PNPLA3 I148M (n (%) CC) | 33 (61) | 32 (63) | 22 (51) |
| Total energy intake (kcal) | 2091.2 ± 636.4 | 2010.5 ± 592.8 | 2078.2 ± 650.5 |
| Carbohydrates (E%) | 44.0 ± 6.0 | 42.6 ± 6.1 | 39.3 ± 7.8 |
| Fiber (g) | 20.5 (10.6) | 19.4 (7.7) | 20.9 (9.0) |
| Protein (E%) | 15.6 (3.9) | 16.2 (4.7) | 17.3 (4.3) |
| Fat (E%) | 35.7 ± 5.6 | 37.1 ± 5.8 | 41.1 ± 7.7 |
| SFA (E%) | 13.2 (2.4) | 13.9 (2.8) | 16.0 (4.9) |
| MUFA (E%) | 13.7 ± 2.7 | 14.1 ± 2.6 | 15.8 ± 3.8 |
| PUFA (E%) | 5.1 (2.6) | 5.0 (2.2) | 5.1 (2.2) |

[a]Data are presented as mean ± standard deviation (SD), median (interquartile range (IQR)) or counts (%). *BMI* Body Mass Index, *DBP* Diastolic Blood Pressure, *HbA1c* Haemoglobin A1c, *HND* Healthy Nordic Diet, *LCPUFA* Low Carbohydrate Polyunsaturated Fatty Acids, *LDL* Low-Density Lipoprotein, *NAFLD* Non-Alcoholic Fatty Liver Disease, *PEth* Phosphatidylethanol, *PNPLA3* Patatin-Like Phospholipase domain-containing protein 3, *SBP* Systolic Blood Pressure, *UC* Usual Care.

Accumulation of liver fat predisposes to non-alcoholic fatty liver disease (NAFLD), a spectrum of diseases strongly associated with impaired glucose and lipid homeostasis seen in type 2 diabetes (T2D)[1]. As T2D develops, the inflammatory state of the steatotic liver is further exacerbated, and may lead to non-alcoholic steatohepatitis (NASH) and progression of fibrotic scarring[2]. Considering a possible causal role of NAFLD in inducing multiple metabolic disorders, targeting liver fat through dietary interventions may be a viable strategy to prevent steatotic liver disease from developing and improve glucose homeostasis in individuals with impaired glucose control or T2D[3].

Short-term trials from our group have demonstrated a reduction in liver fat when individuals consume a diet rich in polyunsaturated fatty acids (PUFA) compared to saturated fatty acids (SFA)[4–6]. Randomized trials that lower total carbohydrates have also shown a

reduction in liver fat[7,8], although effects may differ between studies depending on the contrasting quality of carbohydrates (refined versus whole-grains) and fat (SFA versus PUFA or monounsaturated fatty acids (MUFA))[9–13]. An overall plant-based low-carbohydrate diet with a focus on high-quality carbohydrates, low SFA and high omega-6 PUFA content has not yet been investigated in people with T2D or pre-diabetes. Postprandially, n-6 PUFA are oxidized more rapidly than SFA[14]. In addition, n-6 PUFA have been suggested to inhibit de novo lipogenesis (DNL), a metabolic pathway that is upregulated in T2D and by a higher intake of refined carbohydrates[15]. We hypothesized that a customized "anti-lipogenic diet" including a lowering of carbohydrates and replacing SFA with PUFA, could reduce liver fat content effectively.

For dietary management of T2D, a plant-based, low-fat food pattern has also been recommended, but more long-term clinical studies are highly warranted[16]. A healthy Nordic diet (HND), rich in high-quality carbohydrates (derived from e.g., whole-grain bread and oats) and lower in fat intake with an emphasis of decreasing SFA intake has been shown to improve cardiometabolic risk factors over 6–26 weeks in people with hypercholesterolemia, overweight and metabolic syndrome compared to a habitual average Nordic diet[17–19], but more long-term data in people with T2D are yet lacking[20]. In addition, the effects of a HND on liver fat content is unknown. Thus, examining the effects of an overall plant-based and healthy low-carbohydrate high PUFA (LCPUFA) diet, or a HND, as compared with a usual care (UC) dietary treatment of T2D would be of high clinical relevance and can provide evidence for dietary treatment of the metabolic disorders of T2D. Importantly, there is a need for long-term RCTs investigating diets in people with T2D specifically[16], as well as examining diets for treating NAFLD.

In this work, we investigate the effects of a LCPUFA diet and a HND on liver fat content (primary outcome) and related cardiometabolic risk factors (secondary outcomes) in men and women with T2D or prediabetes during 12 months of intervention. We show that an "anti-lipogenic" LCPUFA diet and a HND causes similar reductions in liver fat and LDL-cholesterol as compared with UC. Only the HND causes further improvements in body weight, glycemic control, liver biochemistry, triglycerides, and inflammation, suggesting the HND as a clinically feasible diet for the management of both T2D and NAFLD.

## Results

Baseline characteristics are presented in Table 1 as median (interquartile range (IQR)), mean ± standard deviation (SD) or n (%). Study participants were 65 (10) years old at baseline, had a BMI of 30.1 ± 3.6 kg/m² (LCPUFA), 29.5 ± 3.7 kg/m² (HND), 30.3 ± 3.2 kg/m² (UC) and a distribution of women/men of 41/59% (LCPUFA), 39/61% (HND) and 37/63% (UC). Median glucose levels were comparable between groups whereas median percentage of liver fat was 6.3 (6.9)% for LCPUFA, 6.4 (5.5)% for HND and 8.7 (10.8)% for UC.

### Drop-out rate and adherence to the assigned diets
Ten out of 150 randomized participants were lost to follow-up after 1 year (< 7%), of which n = 2 dropped out before being informed of their diet (Fig. 1). More participants dropped out in the LCPUFA group than in the other diet groups. Over the follow-up period, 94% of all food bags (1362/1445) were collected at the study site by the participants who remained in the study. Mean differences in energy intake between groups were 178.4 (95% CI: 32.0, 325.0) kcal for LCPUFA vs HND, 142.9 (−11.0, 297.0) kcal for LCPUFA vs UC and −35.5 (−193.0, 122.0) kcal for HND vs UC. Statistically significant differences between groups in the proportion of energy from carbohydrates, fat, SFA, MUFA, PUFA, F&V, including berries, oats, nuts and seeds, sunflower oil and butter were observed. Mean differences in carbohydrate intake were −12.2 (95% CI: −14.8, −9.7) E% for LCPUFA vs HND, −13.2 (−15.9, −10.5) E% for LCPUFA vs UC and −1.0 (−3.8, 1.7) E% for HND vs UC whereas mean differences in fat intake were 13.8 (95% CI: 11.3, 16.2) E% for LCPUFA vs HND, 14.5

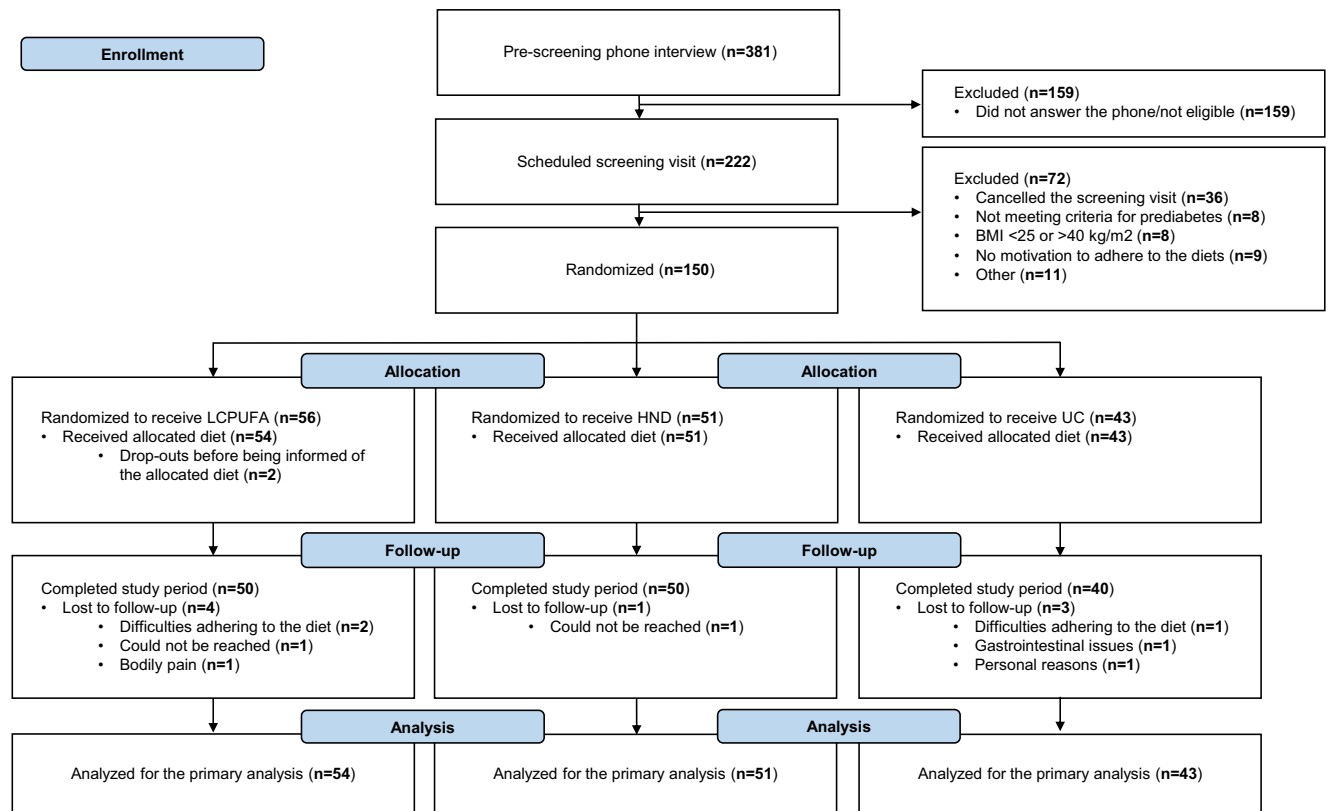

**Fig. 1 | Flow-chart of the NAFLDiet trial.** Follow-up is after 12 months. HND Healthy Nordic Diet, LCPUFA Low-carbohydrate Polyunsaturated Fatty Acids, UC Usual Care. Source data are provided as a Source Data file.

(11.8, 17.1) E% for LCPUFA vs UC and 0.7 (−2.0, 3.4) E% for HND vs UC. The LCPUFA group consumed more PUFA, nuts and seeds and sunflower oil compared to the other groups, whereas the HND group consumed more oats compared to both the LCPUFA and UC group and more whole-grain bread compared to LCPUFA. No statistically significant differences between groups were observed for fiber, protein, fatty fish, red and processed red meat as well as rapeseed oil (Table 2 and Fig. 2). Participants were overall satisfied and motivated to continue with their assigned diet in all diet groups (Supplementary Table 3).

For plasma biomarkers of dietary intake, mean differences in plasma PL linoleic acid between groups were 2.36 (95% CI: 1.49, 3.23)% for LCPUFA vs HND, 2.85 (1.92, 3.78)% for LCPUFA vs UC and 0.49 (−0.45, 1.43)% for HND vs UC. Statistically significant between-group differences were also observed for PL EPA and DHA, as well as for TAG 16:1n-7 (Fig. 3). No statistically significant between-group differences were observed for total AR nor 17:0/21:0 (Supplementary Table 4).

### Primary outcome (liver fat)
A statistically significant between-group effect was observed for liver fat ($P = 0.01$). Median differences between groups were 0.30 (95% CI: −0.52, 1.11)% for LCPUFA vs HND, −1.46 (−2.42, −0.51)% for LCPUFA vs UC, favoring LCPUFA, and −1.76 (−2.96, −0.57)% for HND vs UC, favoring HND (Table 3 and Fig. 4A). Similar estimates were observed in the per-protocol analyses (Supplementary Table 5).

### Secondary outcomes
Statistically significant between-group effects were observed for body weight, HbA1c, CRP, LDL-cholesterol, total cholesterol and triglycerides. Mean differences in body weight between groups were 2.46 (95% CI: 0.71, 4.21) kg for LCPUFA vs HND, −0.31 (−2.18, 1.57) kg for LCPUFA vs UC and −2.77 (−4.64, −0.90) kg for HND vs UC. Differences were also

observed for HbA1c (1.69 (95% CI: 0.24, 3.13) mmol/mol for LCPUFA vs HND, −0.46 (−1.96, 1.04) mmol/mol for LCPUFA vs UC and −2.15 (−3.67, −0.62) mmol/mol for HND vs UC) and CRP (0.26 (95% CI: −0.05, 0.57) mg/L for LCPUFA vs HND, −0.22 (−0.66, 0.22) mg/L for LCPUFA vs UC and −0.48 (−0.84, −0.12) mg/L for HND vs UC) (Table 3 and Fig. 4B–G).

For lipids, between-group differences were observed for LDL-cholesterol (0.02 (95% CI: −0.20, 0.24) mmol/L for LCPUFA vs HND, −0.28 (−0.51, −0.04) mmol/L for LCPUFA vs UC and −0.29 (−0.53, −0.06) mmol/L for HND vs UC) and triglycerides (0.13 (95% CI: −0.12, 0.38) mmol/L for LCPUFA vs HND, −0.24 (−0.50, 0.02) mmol/L for LCPUFA vs UC and −0.37 (−0.62, −0.11) mmol/L for HND vs UC). Similar estimates as for LDL-cholesterol were observed for total cholesterol (Table 3 and Fig. 4B–G). Similar estimates were observed in the per-protocol analyses (Supplementary Table 5).

No differences in HDL-cholesterol, SBP, DBP, FIB-4, fasting plasma glucose, apoB, apoA1, insulin or HOMA-IR were shown (Table 3).

### Exploratory outcomes (liver enzymes)
Statistically significant between-group effects were observed for ALAT and ASAT, but not for GGT. Mean differences in ALAT levels between groups were 0.03 (95% CI: −0.02, 0.07) ukat/L for LCPUFA vs HND, −0.07 (−0.14, 0.01) ukat/L for LCPUFA vs UC and −0.10 (−0.17, −0.02) ukat/L for HND vs UC. Mean differences in ASAT levels between groups were 0.06 (95% CI: 0.00, 0.11) ukat/L for LPUFA vs HND, −0.01 (−0.08, 0.06) ukat/L for LCPUFA vs UC and −0.07 (−0.14, 0.00) ukat/L for HND vs UC (Table 4 and Fig. 5).

### Subgroup analyses
Change in liver fat between groups were in the same direction as for the primary analysis and statistically significant for individuals with T2D (not for prediabetes), those without NAFLD (not NAFLD) as well as

**Table 2 | Within-group changes and overall test of the difference in means/medians between groups of dietary components**

| | LCPUFA (M0)[a] | HND (M0)[a] | UC (M0)[a] | EMM (95% CI)[b] M12-M0 LCPUFA (n = 54) | EMM (95% CI)[b] M12-M0 HND (n = 51) | EMM (95% CI)[b] M12-M0 UC (n = 43) | P-value[c] |
|---|---|---|---|---|---|---|---|
| Total energy (kcal) | 2091.2 ± 636.4 | 2010.5 ± 592.8 | 2078.2 ± 650.5 | −64.2 (−167.0, 38.7) | −242.6 (−348.0, −136.9) | −207.1 (−325.0, −89.2) | 0.04 |
| Carbohydrates (E%) | 44.0 ± 6.0 | 42.6 ± 6.1 | 39.3 ± 7.8 | −10.5 (−12.3, −8.8) | 1.7 (−0.2, 3.5) | 2.7 (0.6, 4.7) | <0.0001 |
| Fiber (g) | 20.5 (10.6) | 19.4 (7.7) | 20.9 (9.0) | 3.1 (1.0, 5.2) | 5.8 (3.7, 7.9) | 6.6 (4.3, 9.0) | 0.05 |
| Protein (E%) | 15.6 (3.9) | 16.2 (4.7) | 17.3 (4.3) | 0.6 (−0.2, 1.4) | 1.4 (0.5, 2.3) | 1.4 (0.5, 2.4) | 0.31 |
| Fat (E%) | 35.7 ± 5.6 | 37.1 ± 5.8 | 41.1 ± 7.7 | 11.1 (9.4, 12.9) | −2.7 (−4.4, −0.9) | −3.3 (−5.4, −1.3) | <0.0001 |
| SFA (E%) | 13.2 (2.4) | 13.9 (2.8) | 16 (4.9) | −2.8 (−3.6, −2.0) | −3.4 (−4.3, −2.6) | −1.5 (−2.5, −0.6) | 0.01 |
| MUFA (E%) | 13.7 ± 2.7 | 14.1 ± 2.6 | 15.8 ± 3.8 | 3.7 (2.8, 4.6) | 0.4 (−0.5, 1.4) | −0.6 (−1.7, 0.5) | <0.0001 |
| PUFA (E%) | 5.1 (2.6) | 5.0 (2.2) | 5.1 (2.2) | 10.0 (8.9, 11.1) | 2.4 (1.3, 3.4) | 0.2 (−1.0, 1.4) | <0.0001 |
| F&V including berries (E%) | 6.0 (5.3) | 5.4 (4.0) | 6.8 (6.7) | 2.3 (0.7, 3.9) | 2.1 (0.6, 3.7) | 6.7 (4.9, 8.5) | 0.0001 |
| Fatty fish (E%) | 1.0 (3.9) | 1.2 (5.1) | 0.0 (4.3) | 1.6 (0.4, 2.8) | 2.3 (1.1, 3.5) | 0.8 (−0.6, 2.2) | 0.31 |
| Red meat and processed red meat (E%) | 8.8 ± 5.8 | 10.1 ± 5.7 | 10.3 ± 7.0 | −2.7 (−4.5, −0.9) | −3.6 (−5.4, −1.9) | −2.4 (−4.3, −0.4) | 0.58 |
| Nuts and seeds (E%) | 0.8 (6.0) | 0.0 (3.0) | 0.0 (5.1) | 16.2 (14.1, 18.4) | 2.7 (0.5, 4.8) | 1.0 (−1.4, 3.4) | <0.0001 |
| Oats (E%) | 0.0 (3.6) | 0.0 (2.5) | 0.0 (2.8) | −1.6 (−2.3, −0.8) | 2.8 (2.0, 3.5) | −0.6 (−1.5, 0.2) | <0.0001 |
| Sunflower oil (E%) | 0.0 (0.0) | 0.0 (0.0) | 0.0 (0.0) | 4.2 (2.8, 5.7) | −2.2 (−2.7, −1.7) | −2.2 (−2.4, −1.9) | <0.0001 |
| Rapeseed oil (E%) | 0.0 (0.0) | 0.0 (0.0) | 0.0 (0.0) | −0.3 (−0.4, −0.2) | −0.2 (−0.3, −0.1) | −0.3 (−0.3, −0.2) | 0.05 |
| Butter (E%) | 1.9 (3.8) | 2.9 (4.4) | 3.3 (5.7) | −2.3 (−2.9, −1.6) | −2.5 (−3.2, −1.8) | −0.3 (−1.1, 0.5) | <0.0001 |
| Whole-grain bread (E%) | 5.4 (8.3) | 5.1 (6.8) | 3.0 (8.4) | −1.6 (−3.1, −0.2) | 2.9 (1.5, 4.3) | 1.8 (0.2, 3.4) | <0.0001 |

[a]Data are presented as mean ± standard deviation (SD) or median (interquartile range (IQR)) for descriptive values at baseline (M0) (n = 146) for each diet.

[b]Estimated marginal means or medians (EMM) with corresponding 95% confidence intervals (CI) are presented for the change in nutrients and/or foods within each diet group. EMMs are conditioned on baseline value of the outcome, presence of type 2 diabetes and gender.

[c]Two-sided p-values are calculated from the general linear model (GLM) or corresponding Kruskal–Wallis (KW) test for the overall test of between-group differences. E% Energy Percentage, EMM Estimated Marginal Means, F&V Fruits and Vegetables, HND Healthy Nordic Diet, LCPUFA Low Carbohydrate Polyunsaturated Fatty Acids, MUFA Monounsaturated Fatty Acids, PUFA Polyunsaturated Fatty Acids, SFA Saturated Fatty Acids, M0 Baseline, M12 Month 12, UC Usual Care.

for individuals with the I148M PNPLA3 CC-genotype (not CG/GG). No sex-specific effects were observed as between-group changes in liver fat was non-statistically significant for both men and women (Supplementary Tables 6 and 7).

Subgroup analyses for HbA1c, total cholesterol, LDL-cholesterol, HDL-cholesterol, triglycerides, apoB and apoA1 are presented in Supplementary Tables 6 and 7. In general, effect estimates were in the same direction as for the primary analysis and between-group statistical significance were generally observed for men (over women), individuals with T2D (over prediabetes), individuals with NAFLD (over no NAFLD) and individuals with the I148M PNPLA3 CC-genotype (over CG/GG). Between-group changes in LDL-cholesterol were only statistically significant for individuals with no NAFLD at baseline (vs NAFLD), whereas HDL-cholesterol was only statistically significant for individuals with the I148M CG/GG genotype (vs CC).

### Sensitivity analyses
Effect estimates from the primary analysis were robust in sensitivity analyses, which included excluding co-randomized household pairs (n = 4 pairs) and using an alternative imputation mechanism whereby values of BMI at months 6 and 12 were utilized as predictors. When excluding co-randomized household pairs, the estimates for liver fat were: LCPUFA vs UC: −1.47 (95% CI: −2.52, −0.42), p = 0.006; HND vs UC: −1.69 (−2.98, −0.41), p = 0.01; LCPUFA vs HND: 0.22 (−0.70, 1.14), p = 0.64. For the latter sensitivity analysis, the estimates for liver fat were as follows: LCPUFA vs UC: −1.51 (95% CI: −2.50, −0.53), p = 0.003; HND vs UC: −1.76 (−2.96, −0.55), p = 0.004; LCPUFA vs HND: 0.24 (−0.56, 1.05), p = 0.55.

### Post-hoc analysis
For liver fat, PM by weight change was 74 (95% CI: −6, 153)% for LCPUFA vs HND, 21 (−38, 80)% for LCPUFA vs UC and 56 (5, 107)% for HND vs UC (Supplementary Fig. 6).

### Descriptive analysis on NAFLD and prediabetes remission
For those with complete data on NAFLD status, 53.6% had remission of NAFLD in the HND group, whereas the corresponding numbers for the LCPUFA diet and UC were 17.4% and 16.7%, respectively. For prediabetes remission, 17.9% had remission in the HND group whereas 17.4% had remission in the LCPUFA group and 7.1% in UC (Supplementary Fig. 4).

### Adverse events
No major differences in the total number of reported adverse events were noticeable between groups. However, the LCPUFA group reported more events of gastrointestinal issues compared to the HND and the UC groups, whereas the UC group reported somewhat more events of bodily pain compared to the LCPUFA and HND groups. Four serious adverse events (n = 2 in the LCPUFA group, n = 1 in the HND group and n = 1 in the UC group) were reported (n = 2 hospitalizations from kidney stones, n = 1 hospitalization from a gallbladder operation and n = 1 hospitalization from a serious allergic reaction) (Supplementary Table 8).

### Co-interventions
Participants were allowed to make changes to their use of medications, but not to their use of supplements, nor alcohol- or physical activity habits. Both allowed and unwanted co-interventions are presented here. Number of participants who engaged in any co-intervention other than changes in dietary supplements, medications, physical activity patterns, or alcohol consumption were somewhat higher in the LCPUFA and UC groups than in the HND group (Supplementary Fig. 2). The number of participants who changed dietary supplements over the study period were higher in the LCPUFA and HND group compared to the UC group. These differences were primarily explained by differences in the initiation of vitamin supplements. No clear differences were noticeable for alcohol consumption between groups (assessed using both WFD and the PEth biomarker) (Supplementary Table 9).

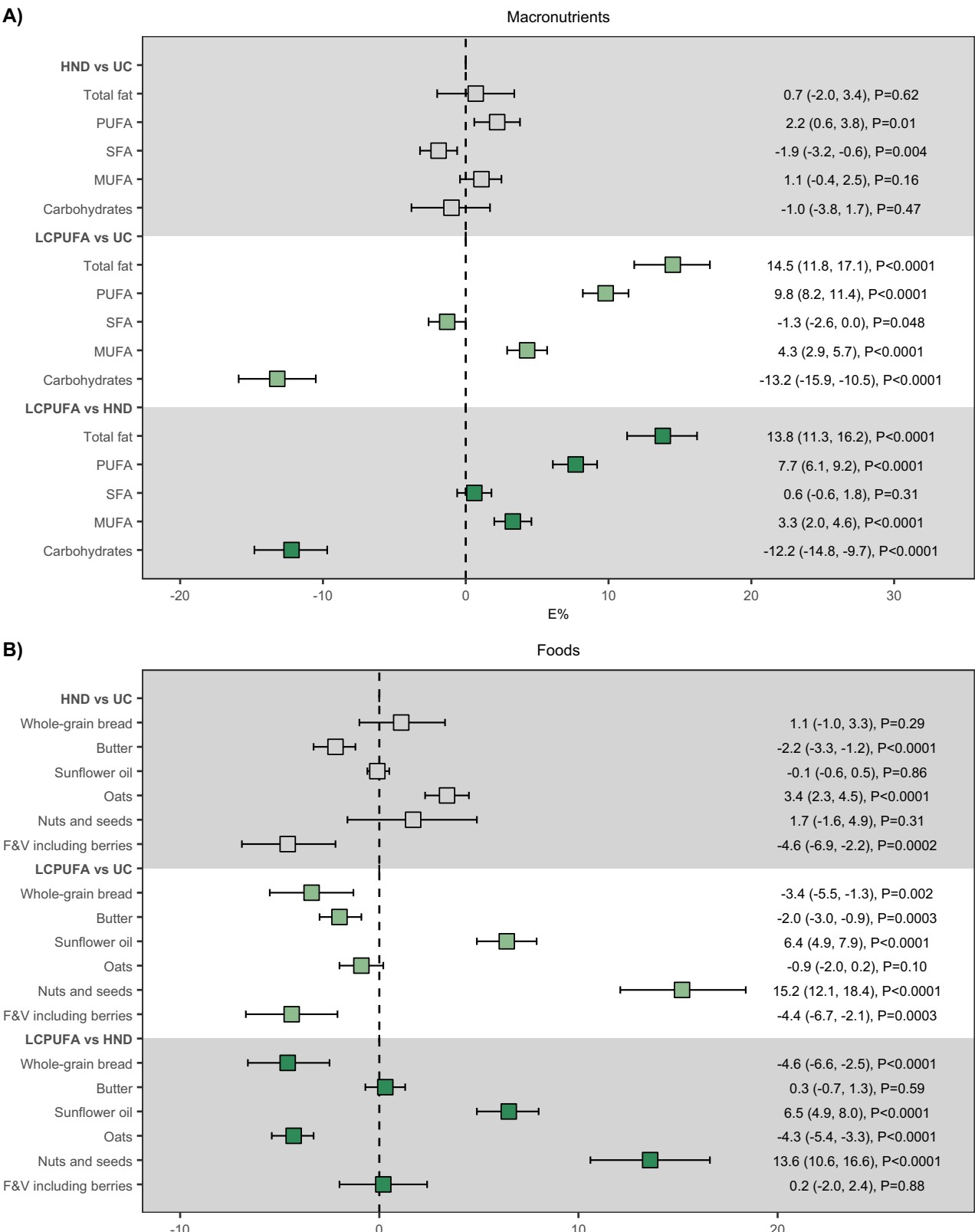

Number of participants who changed total medications relevant to the primary and secondary outcomes of the study (antihypertensives, anticoagulants, lipid-lowering and antidiabetic medications) as well as antidiabetic medications alone, were higher in the HND and UC groups compared to the LCPUFA group. These differences were primarily explained by differences in the increase of metformin dose, initiations of lipid-lowering medications and initiations and change of dose of antihypertensive medications (Supplementary Fig. 3). No data from pedometers were retrieved due to reasons explained in Supplementary Material.

**Fig. 2 | Macronutrients and foods.** Between-group differences of **A** macronutrients (E%) and **B** foods (E%). Intention-to-treat (ITT) point estimates (in E%) and corresponding 95% confidence intervals (CI) within parentheses are demonstrated for macronutrients and foods that were statistically significant in between-group comparisons of the general linear model (GLM)/Kruskal–Wallis (KW) test (Table 2). Estimates and two-sided *p*-values are derived from a GLM or the Willets residual method. E% Energy Percentage, F&V Fruits And Vegetables, HND

Healthy Nordic Diet, LCPUFA Low Carbohydrate Polyunsaturated Fatty Acids, MUFA Monounsaturated Fatty Acids, PUFA Polyunsaturated Fatty Acids, SFA Saturated Fatty Acids, M0 Baseline, M12 Month 12, UC Usual Care. Point estimates with grey color refer to the comparison between HND vs UC; light green LCPUFA vs UC; dark green LCPUFA vs HND. Grey shading is used to separate the three diet comparisons. Source data are provided as a Source Data file. *n* (LCPUFA) = 54; *n* (HND) = 51; *n* (UC) = 43.

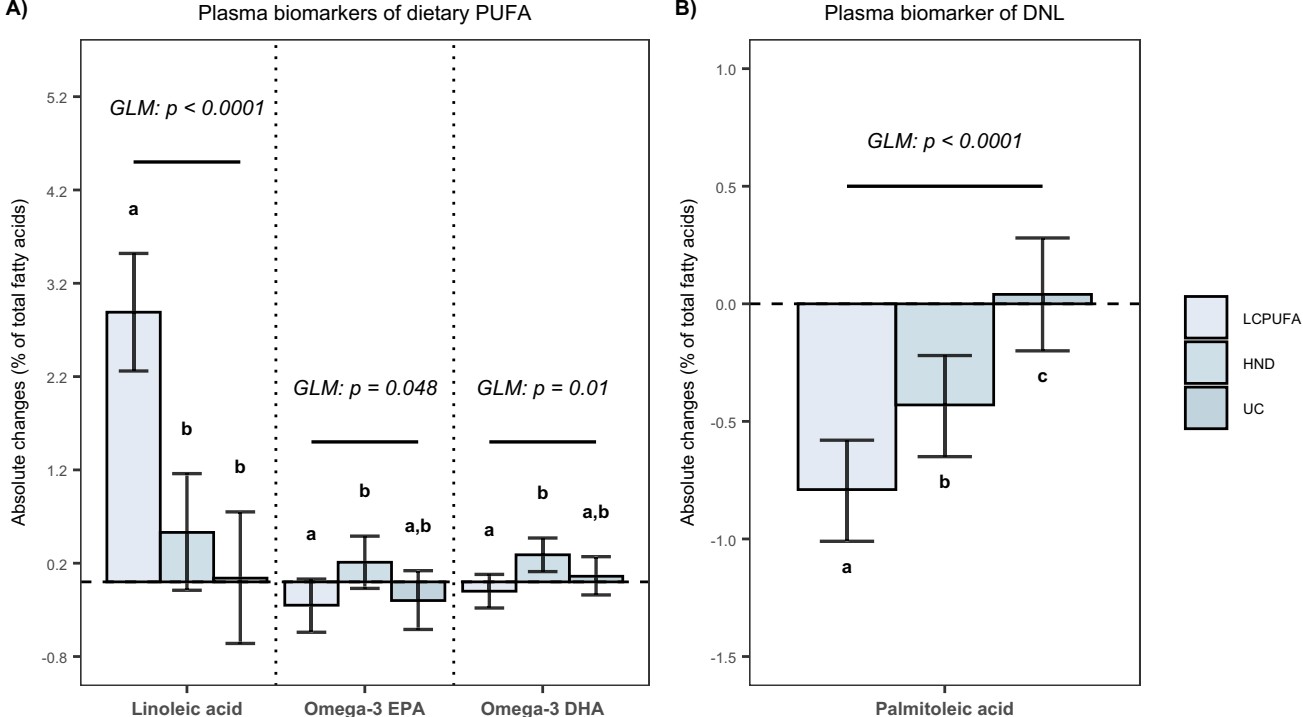

**Fig. 3 | Plasma biomarkers.** Plasma biomarkers of **A** dietary PUFA and **B** DNL. Linoleic acid, omega-3 EPA and omega-3 DHA are measured in phospholipids. Palmitoleic acid is measured in triacylglycerols. Bars represent within-group changes (in % of total fatty acids) with corresponding 95% confidence intervals (CI). The overall test of between-group differences was assessed using a general linear model (GLM). *P*-values are two-sided and no corrections for multiple comparisons have been applied. Estimates are derived from a GLM. Different letters between bars represent a statistically significant between-group effect. DHA

Docosahexaenoic acid, DNL De-Novo Lipogenesis, EPA Eicosapentaenoic acid, HND Healthy Nordic Diet, LCPUFA Low Carbohydrate Polyunsaturated Fatty Acids, PUFA Polyunsaturated Fatty Acids, UC Usual Care. The different shades of blue of the bars represent the different diet groups with the lightest blue referring to LCPUFA, moderate blue to HND and the darkest blue to UC. Source data are provided as a Source Data file, whereby within-group changes in plasma fatty acids and between-group effect estimates with corresponding two-sided p-values are presented. *n* (LCPUFA) = 54; *n* (HND) = 51; *n* (UC) = 43.

## Discussion

In this 1-year randomized trial, we investigated two ad libitum diets that have not been previously tested in individuals with T2D or pre-diabetes. We examined a customized "anti-lipogenic diet", a diet high in plant-based PUFA, but low in carbohydrates and SFA. This LCPUFA diet had beneficial effects on liver fat and plasma LDL-cholesterol concentrations compared with UC, despite no difference in body weight between the groups. We also showed that a healthy low-fat Nordic diet reduced liver fat and LDL-cholesterol to a similar extent as the LCPUFA diet, but notably the HND also reduced CRP, ALAT, ASAT, and triglycerides compared to UC, as well as induced a moderate weight reduction and improvement in long-term glycemic control compared with UC and the LCPUFA diet. Thus, the HND was overall the most effective diet for the treatment of T2D and prediabetes, including improving multiple metabolic disorder seen in NAFLD.

This is to our knowledge the longest study that has investigated the effects of a low-carbohydrate high PUFA diet on liver fat content in T2D and prediabetes. Dalby-Hansen et al. showed in a 6-month long randomized trial in people with T2D that a conventional low-

carbohydrate high-fat (LCHF) diet lowered HbA1c levels and weight compared to a conventional high-carbohydrate, low-fat diet[21]. However, no improvement in NAFLD activity score (NAS), as assessed by liver biopsies, was observed. In addition, the LCHF diet induced higher plasma LDL-cholesterol levels than the low-fat diet, a finding shown in other studies using conventional LCHF diets[16]. It was thus notable that the current overall plant-based low-carbohydrate diet high in PUFA, but low in SFA, clearly reduced LDL-cholesterol as compared with UC. In contrast to our protocol, the low-carbohydrate diet reported by Dalby-Hansen et al. emphasized increased intake of animal-based SFA-rich foods such as butter, meat and high-fat dairy. SFA from butter, in particular, is known to increase LDL-C and liver fat compared to other foods[4,22–24]. These findings emphasize the clinical importance of diet quality and in particular that low-carbohydrate diets should be plant-based with less animal fats to avoid increased LDL-C concentrations and other possible adverse health effects[16]. Interestingly, despite no change in weight between the LCPUFA and the UC group in our study, an absolute change in liver fat content of −1.46% was observed, corroborating previous studies from our group and others that diet may

**Table 3 | Within-group changes and overall test between groups of primary and secondary outcomes**

| | LCPUFA (MO)[a] | HND (MO)[a] | UC (MO)[a] | EMM (95% CI)[b] M12-MO LCPUFA (n = 54) | EMM (95% CI)[b] M12-MO HND (n = 51) | EMM (95% CI)[b] M12-MO UC (n = 43) | P-value[c] |
|---|---|---|---|---|---|---|---|
| Liver fat (%) | 6.3 (6.9) | 6.4 (5.5) | 8.7 (10.8) | −0.61 (−0.90, −0.33) | −0.91 (−1.69, −0.14) | 0.85 (−0.07, 1.76) | 0.01 |
| Weight (kg) | 90.2 ± 13.8 | 87.9 ± 13.6 | 91.2 ± 11.5 | −2.15 (−3.39, −0.91) | −4.61 (−5.85, −3.36) | −1.84 (−3.26, −0.42) | 0.004 |
| Fasting glucose (mmol/L) | 6.7 (1.9) | 6.7 (1.6) | 6.7 (1.0) | −0.36 (−0.59, −0.13) | −0.60 (−0.83, −0.37) | −0.22 (−0.48, 0.03) | 0.08 |
| HbA1c (mmol/L) | 42.0 (12.0) | 40.0 (10.0) | 40.0 (9.0) | 0.36 (−0.64, 1.36) | −1.33 (−2.36, −0.29) | 0.82 (−0.33, 1.97) | 0.01 |
| Total cholesterol (mmol/L) | 4.6 ± 1.2 | 5.1 ± 1.3 | 5.0 ± 1.1 | −0.30 (−0.49, −0.11) | −0.34 (−0.53, −0.15) | 0.03 (−0.19, 0.24) | 0.02 |
| LDL cholesterol (mmol/L) | 2.8 ± 1.1 | 3.2 ± 1.1 | 3.1 ± 1.1 | −0.28 (−0.44, −0.12) | −0.30 (−0.46, −0.14) | −0.01 (−0.19, 0.17) | 0.03 |
| HDL cholesterol (mmol/L) | 1.4 ± 0.4 | 1.4 ± 0.3 | 1.3 ± 0.3 | 0.03 (−0.02, 0.08) | 0.05 (0.00, 0.10) | 0.04 (−0.01, 0.10) | 0.84 |
| Triglycerides (mmol/L) | 1.3 (0.6) | 1.3 (0.8) | 1.4 (0.8) | −0.16 (−0.34, 0.02) | −0.29 (−0.46, −0.12) | 0.08 (−0.11, 0.26) | 0.02 |
| ApoB (g/L) | 0.9 ± 0.3 | 1.0 ± 0.3 | 1.0 ± 0.3 | −0.07 (−0.12, −0.02) | −0.07 (−0.11, −0.02) | −0.02 (−0.07, 0.03) | 0.24 |
| ApoA1 (g/L) | 1.5 ± 0.2 | 1.5 ± 0.2 | 1.5 ± 0.2 | −0.01 (−0.05, 0.03) | −0.02 (−0.06, 0.02) | −0.02 (−0.07, 0.02) | 0.93 |
| Insulin (mE/L) | 14.7 (7.8) | 12.9 (8.8) | 15.2 (11.2) | −1.87 (−3.63, −0.11) | −3.24 (−5.02, −1.46) | −1.55 (−3.64, 0.54) | 0.38 |
| HOMA-IR | 4.8 (2.8) | 3.9 (3.1) | 4.8 (3.6) | −0.79 (−1.46, −0.12) | −1.42 (−2.09, −0.74) | −0.48 (−1.27, 0.31) | 0.17 |
| FIB-4 | 1.4 (0.7) | 1.5 (0.5) | 1.5 (0.5) | 0.24 (0.08, 0.39) | 0.11 (−0.04, 0.25) | 0.07 (−0.10, 0.23) | 0.29 |
| SBP (mmHg) | 130.6 ± 14.6 | 134.3 ± 14.3 | 135.4 ± 13.5 | −2.89 (−6.55, 0.77) | −7.27 (−10.87, −3.67) | −5.82 (−9.91, −1.73) | 0.22 |
| DBP (mmHg) | 83.1 ± 9.0 | 84.8 ± 8.2 | 85.4 ± 8.3 | −3.90 (−5.95, −1.86) | −5.33 (−7.39, −3.28) | −3.92 (−6.24, −1.60) | 0.54 |
| CRP (mg/L) | 1.3 (2.4) | 1.0 (1.4) | 1.6 (2.5) | −0.62 (−0.91, −0.33) | −0.88 (−0.99, −0.76) | −0.40 (−0.74, −0.05) | 0.04 |

[a]Data are presented as mean ± standard deviation (SD) or median (interquartile range (IQR)) for descriptive values at baseline (MO) (n = 142 for liver fat and n = 148 for all other) for each diet.

[b]Estimated marginal means or medians (EMM) with corresponding 95% confidence intervals (CI) are presented for the change in primary and secondary outcomes within each diet group. EMMs are conditioned on baseline value of the outcome, presence of type 2 diabetes and gender.

[c]Two-sided p-values are calculated from the general linear model (GLM) or corresponding Kruskal–Wallis (KW) test for the overall test of between-group differences. ApoA1 Apolipoprotein A1, ApoB Apolipoprotein B, CRP C-Reactive Protein, DBP Diastolic Blood Pressure, EMM Estimated Marginal Means, FIB-4 Fibrosis-4, HbA1c Hemoglobin A1c, HDL High-Density Lipoprotein, HND Healthy Nordic Diet, LCPUFA Low Carbohydrate Polyunsaturated Fatty Acids, LDL Low-Density Lipoprotein, MO Baseline, M12 Month 12, SBP Systolic Blood Pressure, UC Usual Care.

have weight-independent effects on liver fat[4–11,13,22]. In fact, according to our CMA only 21 (95% CI: −38, 80)% of the total causal effect on liver fat could be explained by a change in weight between the LCPUFA and the UC group, although the precision of the estimated mediation effect was low (Supplementary Fig. 6). Furthermore, 16:1n-7 in TAG, a potential biomarker of DNL, was reduced in the LCPUFA diet group compared to both UC and HND, supporting the idea of an anti-lipogenic effect from lowering carbohydrates and increasing dietary PUFA, with the caveat that plasma 16:1n-7 has been questioned lately whether it reflects DNL or not[25].

The HND reduced liver fat by 1.76% compared to UC, of which 56 (95% CI: 5, 107)% of the total causal effect was explained by a change in body weight between groups (Supplementary Fig. 6). Similar reductions in liver fat were observed between the HND and the LCPUFA diet. However, the HND lowered HbA1c and body weight more than the LCPUFA group. The HND also improved several other cardiometabolic risk markers (triglycerides, LDL-C, liver enzymes and CRP) compared to UC, which in part might have been explained by a larger reduction in body weight through higher ad libitum intake of fiber-rich and satiating foods (e.g., oats)[26]. Other potential weight-independent mechanisms that may explain the intervention effects from the HND are increases in the consumption of oats (compared to UC and LCPUFA), whole-grain bread (compared to LCPUFA) and a reduction in butter consumption (compared to UC), dietary changes that may have directly improved postprandial glucose excursions, cholesterol absorption and lipid metabolism[27–29]. Previous short-term randomized trials on the healthy Nordic diet support our findings on the effects on blood lipids and body weight[17–19].

**Clinical relevance**

As NAFLD is defined by the presence of steatosis in more than 5% of liver cells, a 1.46–1.76% absolute difference (corresponding to a relative difference in liver fat of 20–25%) between the LCPUFA group and UC and the HND and UC, respectively, is important. Notably, the liver

fat reduction after HND (−22%), HbA1c and blood lipids were even more marked in participants with T2D (Supplementary Tables 6 and 7), also showing larger differences between diets. Overall, these results are of clinical relevance, especially considering that this effect was sustained at 1 year, and was accompanied by an overall improvement of the cardiometabolic risk profile included in the new MASLD diagnosis, and with few adverse events. Interestingly, Tamaki et al. showed that a 25% reduction in liver fat was accompanied by an increased odds of fibrosis regression in patients with NAFLD over 1.4 years of follow-up[30]. In addition, compared to healthy controls, non-alcoholic fatty liver (NAFL), in the absence of NASH and fibrosis, is associated with all-cause mortality, hepatocellular carcinoma (HCC) and extrahepatic cancers[31,32]. The reduction in liver fat for the intervention diets was furthermore accompanied by a reduction in LDL-cholesterol of 0.28–0.29 mmol/L, corresponding to an estimated 5–10% risk reduction of incident CVD, if sustained long-term[33]. The HND also improved liver enzymes compared to UC, suggesting not only a reduction in liver fat but potentially also other beneficial hepatic effects. The lack of effect on FIB-4 is however, not that surprising, given this fibrosis marker is not very sensitive to smaller changes, and that more substantial interventions over longer time are necessary to see significant changes in fibrosis, especially if the degree of fibrosis is low at baseline, as in the current study.

Notably, the current favorable effects were achieved without advising energy-restricted diets, but was advised merely as ad libitum. Thus, the results of this study should be viewed in terms of this food-based intervention, and may not be comparable to previous more extreme energy-restricted diets that primarily aimed for weight loss.

Notably, our descriptive analysis showed that more than half (~54%) of the randomized patients to the HND had a remission of NAFLD, compared with only ~17% after the other two diets, and the HND group also had the highest rate of remission of prediabetes (Supplementary Fig. 4). Taken together, these results are promising and suggest that a HND may be an effective dietary approach for

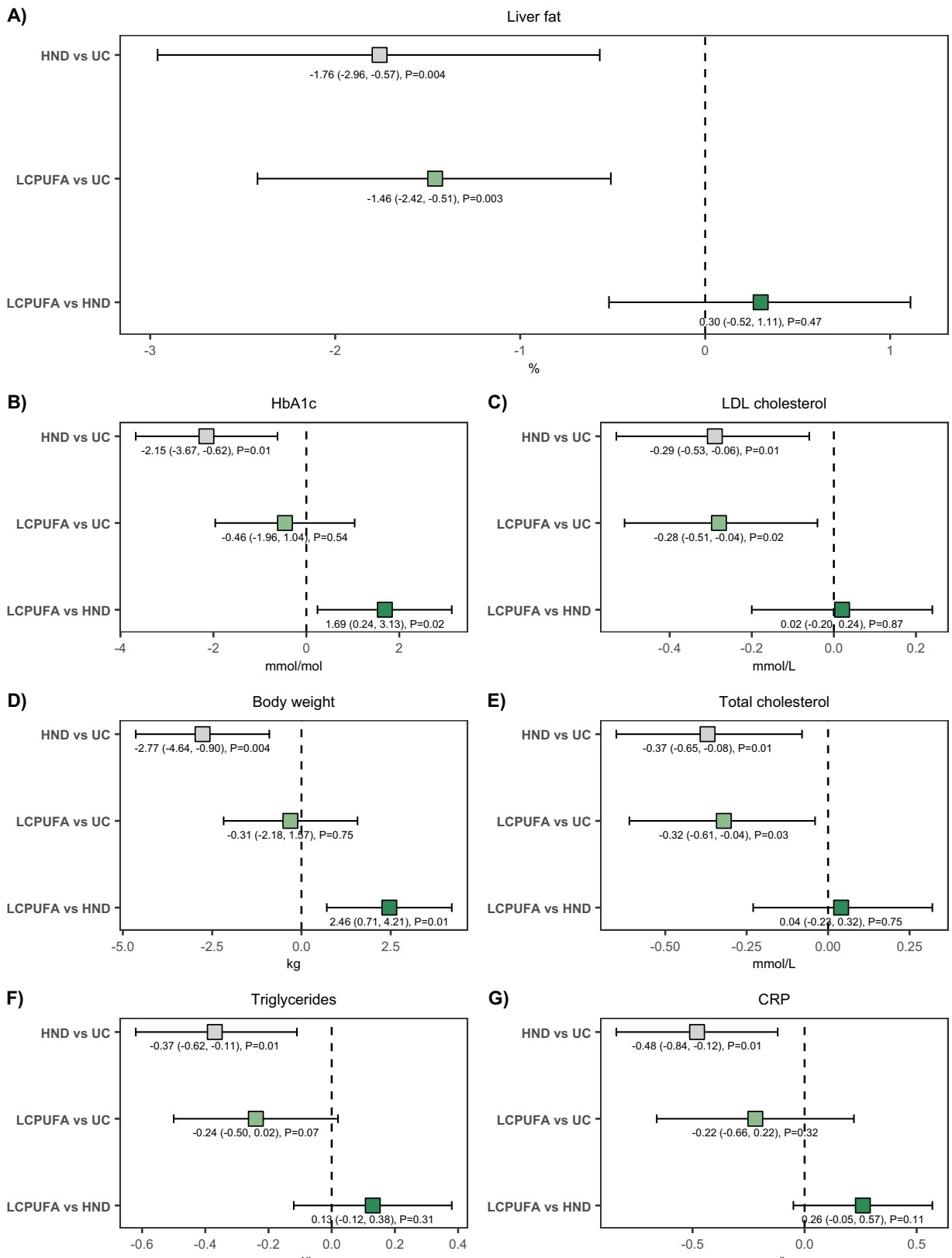

**Fig. 4 | Primary and secondary outcomes.** Between-group differences of **A** primary and **B**–**G** secondary outcomes. Intention-to-treat (ITT) point estimates (in %, kg, mmol/mol, mmol/L, or mg/L) and corresponding 95% confidence intervals (CI) within parentheses are demonstrated for outcomes that were statistically significant in between-group comparisons of the general linear model (GLM)/ Kruskal–Wallis (KW) test (Table 3). Estimates and two-sided *p*-values are derived from a GLM or the Willets residual method. CRP C-Reactive Protein, HbA1c Hemoglobin A1c, HND Healthy Nordic Diet, LCPUFA Low Carbohydrate Poly-unsaturated Fatty Acids, LDL Low-Density Lipoprotein, M0 Baseline, M12 Month 12, UC Usual Care. Point estimates with grey color refer to the comparison between HND vs UC; light green LCPUFA vs UC; dark green LCPUFA vs HND. Source data are provided as a Source Data file. *n* (LCPUFA) = 54; *n* (HND) = 51; *n* (UC) = 43.

**Table 4 | Within-group changes and overall test of the difference in medians between groups of exploratory outcomes**

| | LCPUFA (M0)[a] | HND (M0)[a] | UC (M0)[a] | EMM (95% CI)[b] M12-M0 LCPUFA (n = 54) | EMM (95% CI)[b] M12-M0 HND (n = 51) | EMM (95% CI)[b] M12-M0 UC (n = 43) | P-value[c] |
|---|---|---|---|---|---|---|---|
| ALAT (ukat/L) | 0.39 (0.17) | 0.38 (0.16) | 0.45 (0.25) | −0.05 (−0.09, −0.02) | −0.08 (−0.11, −0.05) | 0.02 (−0.05, 0.08) | 0.02 |
| ASAT (ukat/L) | 0.46 (0.13) | 0.44 (0.15) | 0.44 (0.15) | −0.02 (−0.06, 0.02) | −0.07 (−0.11, −0.03) | 0.00 (−0.06, 0.05) | 0.01 |
| GGT (ukat/L) | 0.40 (0.35) | 0.44 (0.24) | 0.45 (0.36) | −0.03 (−0.06, 0.01) | −0.04 (−0.08, 0.00) | 0.00 (−0.06, 0.06) | 0.39 |

[a]Data are presented as median (interquartile range (IQR)) for descriptive values at baseline (M0) (n = 148) for each diet.
[b]Estimated marginal medians (EMM) with corresponding 95% confidence intervals (CI) are presented for the change in exploratory outcomes within each diet group. EMMs are conditioned on baseline value of the outcome, presence of type 2 diabetes and gender.
[c]Two-sided p-values are calculated from the Kruskal–Wallis (KW) test for the overall test of between-group differences. ALAT Alanine Aminotransferase, ASAT Aspartate Aminotransferase, EMM Estimated Marginal Means, GGT Gamma Glutamyltransferase, HND Healthy Nordic Diet, LCPUFA Low Carbohydrate Polyunsaturated Fatty Acids, UC Usual Care.

treating NAFLD and improving its associated metabolic disorders. Unfortunately, we could not adequately calculate the number of patients that had remission of T2D, partly due to lack of detailed data on medication changes during the study.

In contrast to some[17,19], but not all[18], previous shorter-term trials in non-diabetic populations, the HND did not reduce blood pressure more than the other diets. One possibility for the lack of difference between groups could be that salt restriction was not emphasized as was done in our previous short-term trial[17]. Intake of fruits and vegetables was also higher in the UC group compared to both LCPUFA and HND, which may also explain why the rather pronounced reductions in blood pressure did not significantly differ between the groups.

### Strengths and limitations

There are several strengths and limitations worthy of discussion. First, our primary effect of interest was the ITT effect, and currently not to estimate the effect had everybody in the study adhered to their assigned diet (i.e., the non-naïve per-protocol effect). Although non-adherence to the diets would not bias the ITT effect, it would impact the size of the effect and thus the generalizability of our results[34]. However, WFDs combined with plasma biomarkers from our study indicated good adherence on a group-level, thereby approximating the non-naive per-protocol effect. The per-protocol effect in our study should be interpreted as a complete-case analysis. Secondly, our results may not necessarily be validly extended to other populations with other distributions of potential effect modifiers (such as T2D status, NAFLD status and PNPLA3 allele frequencies, as indicated in our study). Thirdly, data on other potential genetic modifiers closely associated with NAFLD, such as TM6SF2, HSD17B13, and MBOAT7, would have been valuable to have. Lastly, we did not assess histological changes from liver biopsies, which would have provided a valuable complement to the liver fat measurement. However, there are several strengths of this randomized trial that need to be highlighted, including low drop-out rates despite the long intervention period, the use of MRI to assess liver fat content, the blinding of the outcome assessors, good protocol adherence with regards to alcohol consumption (as assessed using both self-reported dietary records and objective biomarkers) and lastly, good overall adherence to the diets. Adherence was excellent with regard to the foods participants received (e.g., 15.2 E% difference in nuts and seeds between LCPUFA and UC and 4.6 E% difference in whole-grain bread between LCPUFA and HND). Differences in levels of LA, the major PUFA in the diet and a biomarker of LA-rich foods, between the LCPUFA group and the other diets clearly supported high dietary adherence and corroborated the findings of self-reported data. However, there were no significant differences between groups in self-reported intake of rapeseed oil or fiber, two key food components in a HND. This might highlight the difficulty in achieving all interventional goals in long-term dietary RCTs, especially those given as oral or written advice, rather than as provided foods. Likewise, we did not observe statistically significant differences in plasma levels of AR between the HND and the other diet groups, which may be explained by the fact that AR in plasma reflect intake of whole-grain from barley and rye, and not oats. Lastly, we did not measure any biomarkers related to the consumption of fruits and vegetables, which would have been a good complement to the self-reported data.

In conclusion, compared with UC, a LCPUFA diet and a HND reduced liver fat and LDL-C to a similar extent, the former outcome possibly mediated by different mechanisms between diet comparisons. Notably, the HND had more favorable effects on HbA1c compared to both the UC and LCPUFA diet, and showed even more pronounced effects on liver fat and metabolic control in the participants with T2D. In addition, only the HND induced significant reductions in triglycerides, inflammation, liver enzymes and body weight compared with UC at 1 year. Notably, these favorable effects were

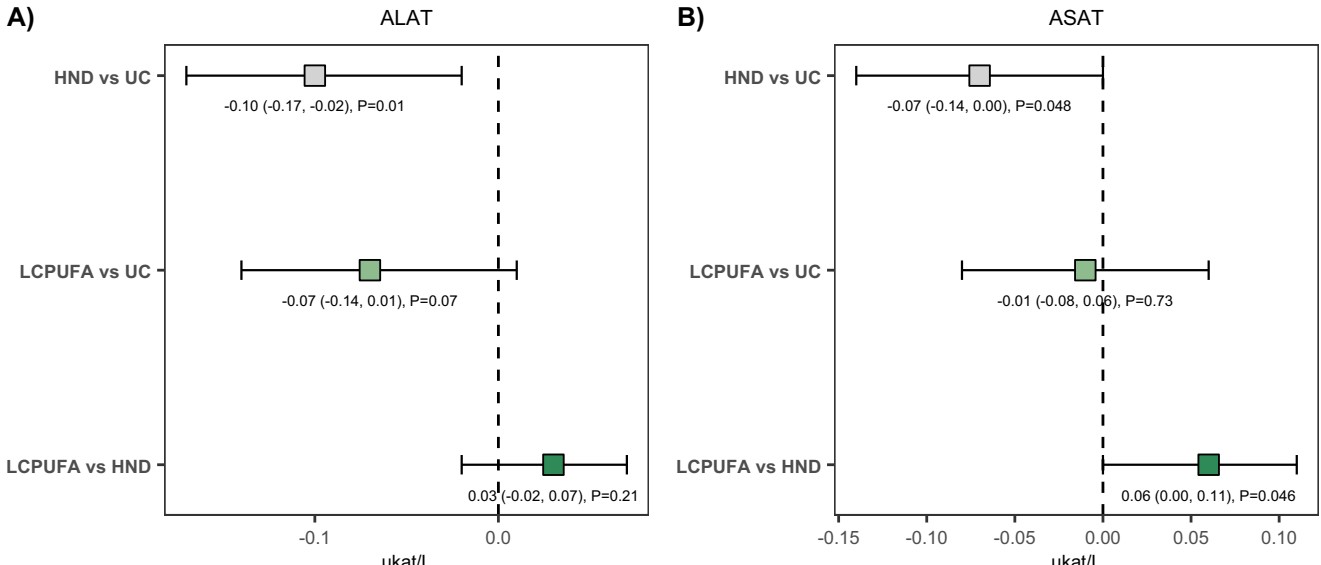

**Fig. 5 | Exploratory outcomes.** Between-group differences of exploratory outcomes **A** ALAT and **B** ASAT. Intention-to-treat (ITT) point estimates (in ukat/L) and corresponding 95% confidence intervals (CI) within parentheses are demonstrated for liver enzyme outcomes that were statistically significant in between-group comparisons of the Kruskal–Wallis (KW) test (Table 4). Estimates and two-sided p-values are derived from the Willets residual method. *ALAT* Alanine

Aminotransferase, *ASAT* Aspartate Aminotransferase, *GGT* Gamma Glutamyl-transferase, *HND* Healthy Nordic Diet, *LCPUFA* Low Carbohydrate Polyunsaturated Fatty Acids, *UC* Usual Care. Point estimates with grey color refer to the comparison between HND vs UC; light green LCPUFA vs UC; dark green LCPUFA vs HND. Source data are provided as a Source Data file. *n* (LCPUFA) = 54; *n* (HND) = 51; *n* (UC) = 43.

achieved without advising energy-restricted diets, but was advised merely as ad libitum. This trial provides clinically relevant evidence suggesting the HND as an effective diet that is well accepted by patients with T2D or prediabetes, and thus could be useful for the long-term management of the multiple cardiometabolic disorders of NAFLD and T2D.

## Methods
The Ethical Review Board of Sweden approved the study and all participants signed a written informed consent prior to inclusion (Dnr: 2019-05111 and 2020-05470).

### Metabolic-dysfunction Associated Steatotic Liver Disease (MASLD)
In June of 2023, a multi-society Delphi consensus statement was released proposing a change in nomenclature from NAFLD to Metabolic-dysfunction Associated Steatotic Liver Disease (MASLD)[35]. This change in nomenclature with its minor modifications to the disease criteria occurred late in the analysis phase of this study. To stay true to the primary name of the trial as well as to further prespecified subgroup analyses specified below, we chose to keep the initial name of NAFLD for this manuscript, with the caveat that nearly 100% of all individuals with NAFLD also meet the criteria for MASLD in Sweden[36]. In the current study, the overlap was 100%.

### Study design
The NAFLDiet trial was a single-site, three-arm parallel-group designed study that randomized participants in a 1:1:1 allocation ratio to follow either a LCPUFA diet, a HND or UC, in line with the Nordic Nutrition Recommendations (NNR) 2012 (ClinicalTrials.gov Identifier: NCT04527965). The NAFLDiet trial was conducted in Uppsala, Sweden between August 2020 and December 2022. The first participant was enrolled 11th of August 2020 and the last participant 15th of December 2021.

Participants were predominantly recruited from a local diabetes register (ANDiU) and a large population-based cohort (EpiHealth), but also through web-based advertisement. Men and women

(self-reported) were eligible to participate if they met the following criteria: 30–75 years of age, Body Mass Index (BMI) 25–40 kg/m², T2D with a duration ≤10 years with no insulin treatment or prediabetes (defined as a fasting plasma glucose ≥5.6 mmol/L or an HbA1c ≥ 39 mmol/mol in accordance with guidelines from American Diabetes Association 2019) and no diagnosed cardiovascular disease (CVD) in the last two years prior to screening. Exclusion criteria included self-reported alcohol intake >20 g/day, contraindications for magnetic resonance imaging (MRI) assessment, unwillingness to follow a new prescribed diet for a year, ≥10% diet-induced weight loss the preceding three months of screening, malignant diseases, severe kidney or liver disease, heart failure or other severe CVD. A total of *n* = 222 individuals were scheduled for a physical screening visit, of which *n* = 150 were included and randomized (Fig. 1).

Randomization (stratified by sex and T2D status) was performed by a researcher not involved in the execution of the study (i.e., no contact with participants and no involvement in primary or secondary outcome ascertainments), using a computerized random-number generator. Knowledge of the allocation sequence was restricted to this researcher. To avoid spill-over effects from the diets, four pairs of participants who lived together were co-randomized. Personnel responsible for assessing study outcomes (including dietary adherence) as well as care providers were blinded to the assigned diets of the participants. Neither the participants nor the study coordinator (M.F.) were blinded to the assigned diets. Participants did not know of their assigned diet until they had completed all measurements from the fasting study baseline visit.

### Diets
At baseline, participants were provided with detailed written and orally presented dietary guidelines tailored to their assigned diet.

### LCPUFA
The LCPUFA group was instructed to limit their intake of carbohydrates to <30 E%, but maintain whole-grain and fiber-rich carbohydrates, and increase their intake of fat to >50 E%, (focusing on increasing omega-6 PUFA and limiting SFA), while keeping protein at

20 E% (focusing on plant-based food sources). To achieve this, ten food-based dietary recommendations were advised, of which four were emphasized; consume at least two table spoons (tbsp.) of sunflower oil per day, two tbsp. of seeds (primarily pumpkin- and sunflower seeds) per day and a handful of nuts $(40 \times g)$ (primarily walnuts, pecan nuts and Brazil nuts) per day. Participants were also guided (through both an information leaflet and a recipe book) on how to replace carbohydrate sources such as pasta, rice, bread and potatoes with other food sources such as bean pasta, cauliflower rice, low-carbohydrate bread rich in seeds and nuts, and root vegetables such as carrots and parsnips.

### HND

The HND group was instructed to limit their intake of fat to 25–30 E% (focusing on PUFA and MUFA) and increase their intake of carbohydrates to 50–55 E% (focusing on increasing whole-grain, fiber-rich carbohydrates and limiting refined carbohydrate sources), while keeping protein at 20 E%. To achieve this, ten food-based dietary recommendations were provided, of which four were emphasized; participants were told to consume at least two portions of whole-grains (primarily oats and rye) per day, two slices of whole-grain bread (with flour made from rye and/or oats) and to focus on foods grown or produced in Sweden and other Nordic countries, such as apples, pears, blueberries, raspberries, cabbage, almonds, salmon, mackerel and herring. Participants were also guided (through both an information leaflet and a recipe book) on how to limit fat intake by replacing full-fat products with low-fat products (e.g., milk, fermented milk and cheese) and considering the amount of salad dressing, cooking oil and bread spread used. The main fat source that was emphasized in the HND group was rapeseed oil.

### UC

The UC group received information on how to follow the current dietary guidelines laid out in the NNR 2012 edition. Protein intake was kept at 20 E%. To partly mask participants from knowing that they had been randomized to the usual care group, extra emphasis was put on increasing fruits and vegetables to 600 grams per day instead of the current recommendation of 500 grams per day. No clear emphasis was placed on using local or Nordic foods in this group. A wider variety of healthy foods were advised, e.g., in contrast to the HND, olive oil could be used instead of rapeseed oil or sunflower oil, and all types of fruit were advised rather than only Nordic types such as apples and pears. The UC group also received an information leaflet and a recipe book.

To enhance diet adherence, study participants received key food items on a monthly to bimonthly basis (once every month the first 7 months followed by once every two months the remaining 5 months). For the LCPUFA group, these foods included sunflower oil, walnuts, cashew nuts, pumpkin seeds and sunflower seeds. For the HND group, these foods included oats, oat bran, oat rice, whole-grain muesli, crisp bread, low-fat margarine (based on rapeseed and sunflower oil), raspberries, beans, lentils and almonds. For the UC group, key food items included crisp bread, whole-grain cereals based on oats, carrots, frozen mango and frozen peas.

All diets emphasized increasing the intake of fruits and vegetables, replacing whole-fat dairy with low-fat dairy and limiting red and processed red meat as well as sugar-sweetened beverages, pastries and other energy-dense sugar/fat-rich snacks. Diets were provided ad libitum; thus, weight loss was not explicitly targeted. More details of the diets can be found in Supplementary Table 1. To mitigate introducing unwanted co-interventions that may influence the size of the effect of both primary and secondary outcomes, participants were instructed to make no major changes in physical activity, alcohol consumption habits or dietary supplements use. However, due to ethical reasons, participants were allowed to make changes to their medications.

### Self-reported and biomarker based dietary adherence and alcohol intake

Dietary adherence was partly assessed using 4 days (3 weekdays and 1 weekend day) weighed food diaries (WFD) at baseline, 6 months and 12 months. Participants were instructed to weigh all their foods and drinks (excluding water) and report the weights in grams. If weighing the food and/or drinks was not feasible, participants were instructed to report the volume of foods and drinks consumed using household measures. The WFD were processed and analyzed in the commercially available software Dietist Net linked to the Swedish National Food Agency (SNFA) food database by a nutritionist not involved in the design or execution of the trial.

Fatty acids in plasma were analyzed and used as objective biomarkers of fat intake. Specifically, the major dietary PUFA, linoleic acid (18:2n-6) in plasma phospholipids (PL) was used as a valid biomarker of PUFA intake from vegetable fats, and omega-3 eicosapentaenoic acid (EPA) (20:5n-3) and omega-3 docosahexaenoic acid (DHA) (22:6n-3) in PL were used as biomarkers of omega-3 PUFA from seafood mainly. Also, palmitoleic acid (16:1n-7) in plasma triacylglycerols (TAG) was used as a marker of carbohydrate-induced DNL[37–39]. Plasma lipids were extracted from fasting blood samples using the Folch method and lipid fractions were subsequently separated using solid phase extraction (SPE)[40]. Fatty acids were methylated and analyzed using gas chromatography (GC), and given as proportions of the total fatty acids analyzed (peak area%).

Plasma alkylresorcinols (AR), biomarkers of whole-grain wheat and rye intake were analyzed by an LC-MS/MS-based method as reported previously[41]. The within and between batch CVs were 15%. Total AR and the ratio between the homologues 17:0 and 21:0 (17:0/21:0) were measured to assess adherence to whole-grain intake from rye and wheat and the proportion of rye to wheat in the diet, respectively[42].

Given the influence of alcohol on liver fat, triglycerides and glycemic control, alcohol intake was assessed by WFD as well as by phosphatidylethanol (PEth), a specific and sensitive biomarker of alcohol intake as measured in plasma at the Uppsala University Hospital clinical chemistry laboratory.

### Primary outcome

For quantification of liver fat, a single breath-hold 6-echo 3D water-fat MRI protocol (IDEAL-IQ) was acquired using a surface coil (UAA). The vendor implemented water-fat and fat fraction reconstructions were used. Scan parameters were: TR = 6.27 ms, TE = 2.45 ms, flip angle 3 deg, nsa 0.73, slice thickness 5 mm, FOV 384 × 384 mm, 90% Phase FOV. Axial orientation was used and in-slice voxel dimensions were 1.5 × 1.5 mm and 5 mm slice thickness was used. Number of slices exported from each scan was 28. Liver mean proton density fat fraction (PDFF) was quantified using an automated segmentation approach. Reference liver segmentations were first created manually by delineations in the water signal images from 26 randomly selected subjects. A UNet++ 2D convolutional neural network was then trained to perform the segmentations using the water-signal images using the Dice loss function[43]. A test set of 8% of the data was set aside. Tenfold cross validation resulted in a segmentation accuracy, in terms of mean Dices scores of 0.981. A mean Dice score of 0.983 was obtained on the test set. The trained model was then applied to all images. Post processing was applied using 3D erosion and filtering of largest connected object in 3D. This to reduce the effect of eventual segmentation errors. All segmentation masks were visually quality controlled. Outlier voxel fat fraction values were also excluded outside ±3 SDs from the liver mean fat fraction. The mean liver proton density fat fraction was uses as the measurement of liver fat content[44]. Outcome assessors were blinded to participants' diet allocations. For subgroup analyses, NAFLD was defined as a liver fat content exceeding 5.6%.

### Secondary and exploratory outcomes

Fasting plasma concentrations of glucose, HbA1c, total cholesterol, LDL-cholesterol, HDL-cholesterol, apoA1, apoB, triglycerides, ASAT, ALAT, GGT, platelet counts and C-reactive protein (CRP) and fasting serum concentrations of insulin were measured by routine laboratory methods at Uppsala University Hospital. FIB-4 was calculated as $(age \times ASAT)/(platelet\ counts \times ALAT)$ and HOMA-IR was calculated as $(insulin \times glucose)/22.5$. Systolic (SBP) and diastolic (DBP) blood pressure were assessed in a sitting position at the research clinic by a specialized research nurse after five minutes of rest using an automated blood pressure device (Omron). The values of three consecutive measurements (allowing for 1 min of rest between) were averaged and reported. Body weight and height were measured at the research clinic after an overnight fast. Height was measured using a stadiometer to the nearest 0.5 cm. Weight was measured in light clothing using bioelectrical impedance analysis (BIA) (Tanita). BMI was calculated as $weight(kg)/height(m)2$.

### Statistical analysis

Sample size calculation was based on Lehr´s formula for the comparison between groups, assuming equal treatment effects for the two experimental diets. A sample size of $n = 37$ in each group was estimated to detect a 2 (SD: ± 3) percentage unit difference in liver fat between the experimental groups and the UC group, with significance level (α) of 0.05 and power (1-β) of 0.80. To achieve the desired power for both the intention-to-treat and the per-protocol analyses, and allowing a 25% dropout rate, we included 50 participants in each group. As NAFLD is diagnosed by a liver fat content exceeding 5.6%, a 2% absolute difference was determined clinically meaningful. A 3% SD was determined based on previous trials in healthy individuals or individuals with overweight or obesity[5,6].

An intention-to-treat (ITT) effect was determined a priori to be the primary effect of interest. All individuals who were randomized and were informed of their assigned diet were included in the ITT-population (Fig. 1). A general linear model (GLM) with diet group as a fixed factor, the outcome as dependent variable (difference between month 12 and baseline) and the following baseline variables as covariates: value of the outcome at baseline, sex (man/woman) and T2D diagnosis (yes/no) were included. Multivariate imputation using chained equations (MICE) ($n = 20$ imputations with predictive mean matching (PMM) as imputation method) was used to account for missing outcome and baseline data, assuming the data was missing at random (MAR). MICE was performed using the mice package in R[45]. The amount of missing data ranged from 5% to 7% for all outcomes except for liver fat that had 14% missing data (Supplementary Table 2). The imputation model included diet group, BMI, age, sex, T2D diagnosis, baseline value of the outcome, and the outcome as variables. Pooling of parameter estimates from each imputed dataset was performed using Rubin´s rules. When the pooled F-parameter estimate of the GLM was statistically significant, linear models for pairwise comparisons were performed. This strategy preserves the type 1 error rate at the nominal level for comparisons of three groups[46]. Between-group differences were expressed as estimated marginal means (EMM) with corresponding 95% confidence intervals (CI). Assumptions of the GLM were checked using residual plots. The Shapiro–Wilk test for residuals was used to examine whether data was normally distributed, with a test statistic $W > 0.95$ indicative of a Gaussian distribution.

When the assumptions of the GLM were not satisfied, data were analyzed using Willett´s residual method[47], which is a non-parametric test for group differences with adjustment for covariates, and was performed as follows. A similar GLM as described above was fitted to the data but with diet group omitted from the model. Residuals from the GLM were saved and subsequently used as the dependent variable

in a Kruskal–Wallis test, with diet group included as factor. Estimated marginal median differences between groups with corresponding 95% CI and p-values were retrieved via bootstrapping ($n = 10,000$ bootstraps, $n = 20$ imputations using MICE) with the use of the bootImpute package in R[48] if the Kruskal–Wallis test was statistically significant.

A per-protocol analysis, including all participants who completed the intervention and who had data on the outcome of interest was determined to be secondary to the ITT-analysis. Similar statistical methods were employed for the estimation of the per-protocol effect except for when the assumptions of the GLM were not satisfied, where the Hodges–Lehman estimator was used to retrieve estimated marginal median differences with corresponding 95% CI.

Prespecified subgroup analyses (by sex, T2D diagnosis, PNPLA3 I148M genotype, and NAFLD status) were performed for the estimation of both the ITT-effect and the PP-effect for the following outcomes: liver fat, HbA1c, LDL-C, total cholesterol, HDL-C, triglycerides, apoB and apoA1. Similar models were specified as for the full population with the exception that the stratifying variable was excluded from the model. Multiple imputation and bootstrapping were performed separately for each subgroup. Due to missing information on baseline liver fat content in $n = 6$ individuals, NAFLD status was first imputed, followed by stratification. The robustness of our findings was explored in sensitivity analyses by excluding participants who were co-randomized as pairs ($n = 4$ pairs) and by including values of BMI at month 6 and month 12 in the MICE model. Sensitivity analyses were not prespecified in the statistical analysis plan (SAP) (Supplementary Note 1).

Lastly, as an additional post-hoc analysis, a causal mediation analysis (CMA) for our primary outcome was conducted using the regmedint package in R with weight change from 0 to 12 months included as the mediator[49–51]. Proportion mediated (PM) by weight change was calculated by dividing the natural indirect effect (NIE) by the total causal effect (TE). A detailed description of the methodology underlying the CMA can be found in Supplementary Material.

The original SAP can be found at ClinicalTrials.gov (ClinicalTrials.gov Identifier: NCT04527965). We have modified the statistical analysis compared to the SAP to allow comparisons between all three treatment groups as described above. Other minor modifications to the SAP are described in supplementary material. A *p*-value < 0.05 (two-sided) was determined to be statistically significant. R (R Core Team, Vienna, Austria) version 4.2.3 and IBM SPSS Statistics version 28.0.1.0 (142) were used to analyze the data.

### Reporting summary

Further information on research design is available in the Nature Portfolio Reporting Summary linked to this article.

## Data availability

All relevant data supporting the findings of this study are available within the main manuscript, the Supplementary Material, or as Source Data. Because the study includes sensitive patient data, the pseudonymized individual-level data from the NAFLDiet study cannot be made publicly available. Access may be granted upon request to the corresponding author, but requires a clear project plan and prior approval from the Swedish Ethical Review Authority. The data will be shared exclusively for research purposes. Inquiries will be answered within approximately one month. The study protocol, statistical analysis plan (SAP), and CONSORT checklist are available in the Supplementary Material under Supplementary Note 1 and 2. Source data are provided with this paper.

## Code availability

The code for the analyses is available from the corresponding author upon reasonable request.

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

## Acknowledgements

The NAFLDiet study was supported by the Heart-Lung Foundation (Dnr: 20180709) (U.U.), Swedish research council FORMAS (Dnr: FR-2019/0007) (U.U.), Swedish Diabetes Foundation (Dnr: 2018-389) (U.U.), and Excellence of Diabetes Research in Sweden (EXODIAB) (U.U.). We thank Lantmännen Research Foundation for donating some of the study foods. Funders had no role in the design, analysis or interpretation of the study. We would like to thank Kerstin Marttala, Sara Hansson, Lillemor Källström, Andréa Flores, Gunilla Arvidsson, Anders Lundberg, Marie Åhman, Hernani Vieira, Anna Thörnblom and colleagues at the MRI unit at Uppsala University Hospital for their contribution to this study.

## Author contributions

U.R. conceptualized the study and led its overall design. M.F., F.R., J.K., L.B., J.V., F.Ror., J.S., L.L., R.L., M.O-M., and U.R. helped design the research. M.F., F.R., J.K., M.M., P-O.C., J.H., L.J., N.A., H-E.J., R.L., M.O-H., and H.A. contributed to data collection and/or conducted the research. M.F. and L.B. performed the statistical analyses. M.F. drafted the initial version of the manuscript, and all authors contributed to critical revisions and approved the final manuscript. U.R. had primary responsibility for the final content.

## Funding

## Competing interests

J.V. reports advisory board for Novo Nordisk, and lecture fees from Gore Medical and Norgine. J.S. reports direct or indirect stock ownership in companies (Anagram kommunikation AB, Sence Research AB, Symptoms Europe AB, MinForskning AB), providing services to companies and authorities in the health sector, including Amgen, AstraZeneca, Bayer, Boehringer, Eli Lilly, Gilead, GSK, Göteborg University, Itrim, Ipsen, Janssen, Karolinska Institutet, LIF, Linköping University, Novo Nordisk, Parexel, Pfizer, Region Stockholm, Region Uppsala, Sanofi, STRAMA, Takeda, TLV, Uppsala University, Vifor Pharma, WeMind. J.K. reports being co-founder, stock owner and part time employee at Antaros Medical AB. No other competing interests are reported.

## Additional information

**Michael Fridén**[1], **Fredrik Rosqvist**[1,2], **Joel Kullberg**[3,4], **Lars Berglund** ⓘ [1,5,6], **Johan Vessby**[7], **Mats Martinell**[1], **Per-Ola Carlsson**[8], **Johannes Hulthe**[4], **Lars Johansson** ⓘ [4], **Nouman Ahmad**[3], **Hans-Erik Johansson**[1], **Fredrik Rorsman**[7], **Johan Sundström** ⓘ [7], **Lars Lind**[7], **Rikard Landberg** ⓘ [9,10], **Marju Orho-Melander** ⓘ [11], **Håkan Ahlström**[3] & **Ulf Risérus** ⓘ [1] ✉

[1]Department of Public Health and Caring Sciences, Uppsala University, Uppsala, Sweden. [2]Department of Food Studies, Nutrition and Dietetics, Uppsala University, Uppsala, Sweden. [3]Department of Surgical Sciences, Radiology, Uppsala University, Uppsala, Sweden. [4]Antaros Medical AB, Gothenburg, Sweden. [5]School of Health and Welfare, Dalarna University, Falun, Sweden. [6]Epistat AB, Uppsala, Sweden. [7]Department of Medical Sciences, Uppsala University, Uppsala, Sweden. [8]Department of Medical Cell Biology, Uppsala University, Uppsala, Sweden. [9]Division of Food and Nutrition Science, Department of Life Sciences, Chalmers University of Technology, Gothenburg, Sweden. [10]Wallenberg Laboratory and Department of Molecular and Clinical Medicine, Institute of Medicine, Sahlgrenska Academy, University of Gothenburg, Gothenburg, Sweden. [11]Department of Clinical Sciences Malmö, Lund University, Lund, Sweden. ✉e-mail: ulf.riserus@uu.se

