## [Transparent Peer Review file · Nature Communications]

Effects of an anti-lipogenic low-carbohydrate high polyunsaturated fat diet or a healthy Nordic diet versus usual care on liver fat and cardiometabolic disorders in type 2 diabetes or prediabetes: a randomized controlled trial (NAFLDiet)

Corresponding Author: Professor Ulf Risérus

Version 0:

Reviewer comments:

Reviewer #2

(Remarks to the Author)

All comments have been adequately addressed and the analysis methods used are reasonable and supported.

Reviewer #3

(Remarks to the Author)

Dear Authors,

I commend the authors for a great job and I think this will be a landmark paper in the field of nutrition. Authors have responded to many queries. However, I have a couple of comments. As authors know human genetics contribute strongly to MASLD onset and progression. I urge authors to genotype at least PNPLA3 and HSD17B13 variants and possibly for TM6SF2 and MBOAT7. I understand that this was not in the primary aim of the study but human genetics is randomized at birth and therefore results will be valuable even if p values are nominal. Most importantly if there is an interaction it could also strengthen the therapeutical effect of the intervention.

I may also have missed this but please use phosphatidylethanol to reclassify participants into MASLD and MetALD it is this of interest in the field.

Reviewer #4

(Remarks to the Author)

No further comments.

Reviewer #5

(Remarks to the Author)

Response to reviewers from Nature Communications

We would like to sincerely thank the editor and all reviewers for reading and providing constructive feedback to improve this manuscript. We have addressed the comments from Reviewer #3 below.

Sincerely,

Prof. Ulf Risérus

Reviewer #2 (Remarks to the Author):

All comments have been adequately addressed and the analysis methods used are reasonable and supported.

Response: We thank you for reviewing this manuscript.

Reviewer #3 (Remarks to the Author):

Dear Authors,

I commend the authors for a great job and I think this will be a landmark paper in the field of nutrition. Authors have responded to many queries. However, I have a couple of comments. As authors know human genetics contribute strongly to MASLD onset and progression. I urge authors to genotype at least PNPLA3 and HSD17B13 variants and possibly for TM6SF2 and MBOAT7. I understand that this was not in the primary aim of the study but human genetics is randomized at birth and therefore results will be valuable even if p values are nominal. Most importantly if there is an interaction it could also strengthen the therapeutic effect of the intervention.

I may also have missed this but please use phosphatidylethanol to reclassify participants into MASLD and MetALD it is this of interest in the field.

Response: We thank you for reviewing this manuscript. We do acknowledge that human genetics play a significant role in the development (and progression) of MASLD. We therefore prespecified subgroup analyses based on a common PNPLA3 gene variant (PNPLA3 I148M) in the statistical analysis plan (SAP). Results from these analyses are shown in Supplementary Table 6 and Supplementary Table 7. We also show the distribution of these PNPLA3 alleles (CC vs CG/GG) in individuals with or without NAFLD at baseline in Supplementary Table 9. Genotyping for other genetic variants associated with MASLD, such as HSD17B13, TM6SF2 and MBOAT7, and investigating any subgroup-specific effects across these gene variants, is a good (and important) suggestion for future post-hoc analyses of this trial. We added a sentence on this on line 599-601, under strengths and limitations.

With regards to the definition of MASLD (we refer to it as NAFLD in the paper due to reasons outlined under the subheading “**Metabolic-dysfunction Associated Steatotic Liver Disease (MASLD)**“), we excluded individuals with >20 grams of self-reported alcohol intake for women and >30 grams for men at baseline. Therefore, no participant was deemed an over-consumer of alcohol. These cut-off values for self-reported alcohol intake are based on several consensus statements and current guidelines. As there are currently no agreement on the level of Peth-value a patients has to have to be defined as MASLD or MetALD, we chose not to include this in the paper.

For the interest of the reviewer however, using an arbitrary cut-off of 0.30 $\mu\text{mol/L}$ as suggested by some authors (but not all), only n=6 (of which all had >5.6% liver fat) had PEth-values over this cut-off at baseline and n=5 (of which all had >5.6% liver fat) had PEth-values exceeding this cut-off at month 12. Due to the low numbers of potential misclassifications in conjunction with no consensus agreement of an established PEth cut-off value in practice, we chose to stay with our original definition of NAFLD (later called MASLD) outlined in the protocol and the SAP.

Interestingly however, as stated in the paper, we had 100 % overlap between the older definition of NAFLD with the newer definition of MASLD, including cardiometabolic risk markers.

Reviewer #4 (Remarks to the Author):

No further comments.

Response: We thank you for reviewing this manuscript.